# Macroscopic and Microscopic Characteristics of Strength Degradation of Silty Soil Improved by Regenerated Polyester Fibers under Dry–Wet Cycling

**DOI:** 10.3390/polym15224367

**Published:** 2023-11-09

**Authors:** Xiaoyan Liu, Meng Han, Tong Liu, Lulu Liu

**Affiliations:** 1School of Mechanics and Civil Engineering, China University of Mining and Technology, Xuzhou 221116, China; luckyliuxiaoyan@cumt.edu.cn; 2School of Civil Engineering, Dalian University of Technology, Dalian 116024, China; hanmeng320@mail.dlut.edu.cn; 3School of Civil Engineering, Sun Yat-Sen University, Guangzhou 510275, China; liut339@mail2.sysu.edu.cn; 4State Key Laboratory for Geomechanics and Deep Underground Engineering, China University of Mining and Technology, Xuzhou 221116, China

**Keywords:** dry–wet cycling, reclaimed polyester fiber, silt subgrade, macroscale and microscale effects, shear strength

## Abstract

The structural stability of silt foundations, particularly sensitive to moisture content, can be severely compromised by recurring wetting and drying processes. This not only threatens the foundational integrity but also raises grave concerns about the long-term safety of major civil engineering endeavors. Addressing this critical issue, our study delves into the transformative effects of reclaimed polyester fiber on subgrade silt exposed to such environmental stressors. Through rigorous wet–dry cycle tests on this enhanced soil, we evaluate shifts in shear strength across varying confining pressures. We also dissect the interplay between average pore diameter, particle distribution, and morphology in influencing the soil’s microstructural responses to these cycles. A detailed analysis traces the structural damage timeline in the treated soil, elucidating the intertwined micro–macro dynamics driving strength reduction. Key discoveries indicate a notably non-linear trajectory of shear strength degradation, marked by distinct phases of rapid, subdued, and stabilized strength attrition. Alterations within the micropores induce a rise in both their count and size, ultimately diminishing the total volume proportion of the reinforced soil. Intriguingly, particle distribution is directly tied to the wet–dry cycle frequency, while the fractal dimension of soil particles consistently wanes. This research identifies cement hydrolysis and pore expansion as the dominant culprits behind the observed macroscopic strength degradation due to incessant wet–dry cycles. These revelations hold profound implications for risk management and infrastructural strategizing in areas dominated by silt foundations.

## 1. Introduction

With the advancement of China’s Belt and Road Initiative, coupled with the national strategy to bolster its transportation infrastructure, there has been a marked escalation in transportation engineering projects across the country. Particularly in the southeastern coastal and inland regions, engineers frequently encounter suboptimal subgrades like silt during roadbed construction [1,2,3]. Characterized by susceptibility to dust dispersion in arid conditions, liquefaction upon water contact, high capillarity, and poor compatibility, silt poses considerable challenges such as deformation, cracking, and instability of the road surface, thereby complicating roadbed construction [4,5]. Current remedial measures involve the utilization of inorganic materials—namely, cement, fly ash, and lime—to enhance the mechanical properties, frost resistance, and hydrological stability of silt-based roadbeds [6,7]. Nonetheless, these reinforced structures exhibit persistent vulnerabilities, including inadequate crack resistance, susceptibility to liquefaction, and environmental detriment [8,9]. Consequently, investigating eco-friendly and high-performance materials for silt roadbed reinforcement is a research endeavor of substantial importance.

Traditionally, soil reinforcement predominantly employs layered or strip materials, including geogrids [10], geotextiles [11,12], and other geosynthetic entities. While these materials augment the soil’s mechanical attributes and notably mitigate local vertical settlement, they occasionally engender a frail interface between the steel reinforcement and the soil. This vulnerability can lead to deformation or even failure of the reinforced domain. In response, an innovative fiber reinforcement approach has been introduced, characterized by discrete random reinforcement. This method boasts superior strength and commendable dispersibility. Fiber reinforcement entails the homogenous integration of dispersed fibers into soil, enhancing its mechanical characteristics. It stands as an efficient, eco-friendly solution for stabilizing fine-grained soil. Current research largely revolves around glass fiber [13], polypropylene fiber [14,15,16], polyethylene fiber [17,18,19], and polyester fiber [20]. While regenerated polyester fibers find their primary application in enhancing asphalt concrete or concrete pavements—thereby curbing shrinkage cracks and bolstering soil mechanics—explorations into their potential in fortifying fine-grained soil for road construction remain limited and warrant further investigation.

The macroscopic attributes and the microscopic composition of enhanced clayey soil are intrinsically linked. Since the advent of the 20th century, the notion of a clay honeycomb structure, introduced by the pioneers of soil mechanics, has laid a foundational framework for microscopic soil structure research. Early in this domain, Terzaghi acknowledged the pivotal role of soil structure in shaping its strength parameters and introduced the idea of soil microstructure [21]. Subsequent studies by Yang et al. [22] analyzed the role of particle morphology in the structural stability of powdered soil. Delage et al. [23] undertook both qualitative and quantitative evaluations of compacted silt’s microstructure. Building on this, Wood et al. [24] delved into the impacts of sedimentation patterns and clay proportion on silt particle contacts, crafting a methodology to quantitatively delineate the microstructure based on grain contact stability. Research by Monroy et al. [25], Mašín [26], and Horpibulsuk et al. [27] probed into the microscopic make-up of various soil types, including unsaturated flaky clay and the silt from the Yellow River delta. Their work drew connections between silt’s anisotropy and composition and its microstructural properties. While these studies provide a strong base concerning the interplay between the macroscopic mechanical attributes and silty soil microstructure, the properties of silty soils are not consistent across regions. Presently, there is a research gap regarding the multiscale interactions between the microstructural degradation and macroscopic mechanical responses in regenerated polyester-fiber-reinforced silty soil, signaling the need for further inquiry.

Groundwater-level fluctuations, combined with surface evaporation and precipitation, lead to recurrent wet–dry cycles. These cycles adversely affect the microstructural integrity of both rock and soil, subsequently reducing their macroscopic strength. Such degradation significantly threatens the long-term stability and safety of foundational structures, especially when constructed from water-sensitive materials like silty clay [28]. Various studies have elucidated the influencing parameters on soil mechanics. For instance, Brandon et al. [29] demonstrated that static pressure and vibration significantly impact the density of field silt. Phan et al. [30] employed drained and undrained triaxial tests on sandy soils to reveal that liquefaction resistance improves with increased rolling resistance of fines [31,32]. Thevanayagam and Martin [33] performed field liquefaction and laboratory permeability tests on silt, indicating that the water content and compaction degree affect both the collapse rate and permeability coefficient. Additionally, Ren and Hu [34] investigated the liquid limit of silty clay, highlighting the non-linear relationship between the liquid limit and plasticity index. Capillary water absorption is another crucial mechanism affecting moisture levels in subgrade silt. Dardouri and Sghaier [35] explored this by establishing a functional relationship between capillary water rise height and the permeability coefficient of sandy soil. Wu and Yu [36] employed micro-CT scans and numerical simulations to understand the seepage process and humidity distribution in vertical silty soil. Despite the extant literature, inconsistencies in methodological approaches and results, coupled with the scarcity of comprehensive studies on the macro–micro degradation of enhanced silty soil under wet–dry cycles, necessitate further research in this domain.

This chapter offers an in-depth exploration of the multiscale mechanisms that dictate the relationship between microstructural damage and macroscopic mechanical behavior in polyester fiber-reinforced soil exposed to dry-wet cycles. The influence of these cycles on soil strength was assessed using tests performed on specimens mixed optimally, with dry density and cycling paths as variables. Microstructural changes were scrutinized using Scanning Electron Microscopy (SEM) (This instrument is produced by Nanjing Nanguang Geological Instrument Co., Ltd. in Nanjing, China) and digital image processing methods. This chapter then delineates the progressive nature of structural damage in the reinforced soil, highlighting the underlying mechanisms that drive strength degradation on both micro and macro levels. These insights provide a foundational reference for subsequent studies examining the effects of dry–wet cycles in fiber-reinforced soil.

## 2. Experimental Study

### 2.1. Materials

#### 2.1.1. Silty Soil

The soil utilized in the experiment was sourced from a highway construction site in Dongtai City, Yancheng, Jiangsu Province, as depicted in Figure 1. Figure 2 illustrates the particle size distribution curve for the silt, revealing the respective percentages of clay particles (d < 5 μm), silt particles (5 μm < d < 75 μm), and sand particles (75 μm < d < 200 μm) as 11.3%, 79.8%, and 8.9%. Laboratory tests were conducted in compliance with the Geotechnical Test Code (SL 237-1999), and the soil’s fundamental physical properties were determined, as presented in Table 1. The soil sample’s liquid limit (wL) is below 50%, and its plasticity index (IP) is under 10, categorizing it as low-liquid-limit silt. The plastic limit is denoted by PL. T Table 2 provides the silt’s chemical composition, predominantly comprising SiO_2_ (61.32%) and Al_2_O_3_ (13.24%), with trace amounts of CaO, Fe_2_O_3_, and K_2_O. The optimum moisture content of soil is determined using the standard compaction test. First, a series of soil samples are prepared, each with varying water content. Using standardized compaction equipment and procedures, each sample is compacted to a specified number of layers, and the dry density of each is measured. Finally, a dry density curve is plotted based on the moisture content. The moisture content corresponding to the maximum dry density from the curve is identified as the optimum moisture content.

#### 2.1.2. Renewable Polyester Fiber

The renewable polyester fiber (RP fiber) used in this study was obtained from a textile factory in Tai’an. Its density ranges from 1.31 to 1.37 g/cm^3^ and its tensile strength, measured in single filament bundles, varies between 200 and 400 MPa. The breaking elongation is within the range of 140.6% to 154.7%, as depicted in Figure 3.

#### 2.1.3. Lime, Fly Ash, Gypsum

Table 3 delineates the fundamental physical properties of the quicklime utilized in this research. An X-ray fluorescence spectrometer was employed to assess the lime content and chemical composition, the results of which are articulated in Table 4. The combined content of CaO and MgO was ascertained to be 65.69%, categorizing the quicklime as Class III calcium lime. In this investigation, lime served as an alkaline activator to establish an alkaline milieu.

Fly ash, a byproduct extracted from the emissions of coal-fired boilers, predominantly consists of alumina, silica, and iron oxide, with particulates generally in the micrometer range. This material holds potential as a liner for waste containers, particularly when amalgamated with specific minerals such as lime and bentonite. When utilized for soil stabilization, fly ash offers cost advantages and superior freeze–thaw resistance in comparison to cement [37]. The principal chemical composition and performance metrics of the fly ash are elucidated in Table 5.

Renewable building gypsum, derived from re-grinding and calcining dehydrated primary building gypsum, undergoes a more frequent transition between the hemihydrate and dihydrate phases than its primary equivalent. However, both variants predominantly consist of the mineral calcium sulfate hemihydrate. Impressively, the mechanical properties of waste gypsum are on par with, if not superior to, those of primary gypsum. This not only results in lowered material expenses and enhanced land conservation but also advances environmental protection. Moreover, it encourages the prudent utilization of natural gypsum mineral resources. Consequently, the adoption of renewable building gypsum offers considerable advantages in terms of sustainability, environmental impact, and economic efficiency. For this study, renewable building gypsum powder, primarily consisting of calcium sulfate hemihydrate (CaSO_4_·1/2H_2_O), was employed.

### 2.2. Test Methods

To examine the multiscale impact of reclaimed polyester fiber on enhanced silty soil and to assess the influence of dry–wet cycles, a comprehensive suite of tests encompassing dry–wet cycle evaluations and microstructural analyses of pore and particle configurations was conducted.

#### 2.2.1. Dry–Wet Cycle Test

According to the testing standards [38], the soil samples were meticulously prepared to ensure accurate testing results. Initially, they were dried at a consistent temperature of 105 °C to remove any moisture. Once dried, the samples were crushed using a mechanical crusher to break down the larger particles and then sieved through a 2 mm mesh to maintain a uniform particle size distribution. This meticulous sieving process ensured that only particles smaller than 2 mm were utilized in subsequent procedures. After sieving, the soil was layered in a controlled manner, and moisture was incrementally added to achieve the optimal moisture content, ensuring that the soil had the best consistency for testing. This was followed by a 24 h settling period, allowing the moisture to evenly disperse throughout the sample, guaranteeing uniformity. To enhance the soil’s properties and meet the research objectives, a specialized mixture of regenerated polyester fibers, quicklime, fly ash, and gypsum was prepared. Each ingredient was measured and mixed in specific proportions, as dictated by the research protocol, to achieve the desired characteristics of the modified soil. Once the mixture was ready, standard triaxial specimens were fabricated. These specimens, measuring 39.1 mm in diameter and 80 mm in height, were shaped using a static pressure molding technique. This process was executed with precision to ensure that each specimen met the target dry density specifications, ensuring consistent and reproducible test results. Understanding the importance of moisture distribution within the specimen, a final conditioning step was introduced. Before subjecting the specimens to the wet–dry cycle testing, they were hermetically sealed to prevent external moisture intrusion. They were then stored in a climate-controlled chamber, maintaining a temperature of 20 °C and a relative humidity of 98%, for 48 h. This conditioning ensured that water was uniformly distributed throughout the specimen, creating an ideal environment for the subsequent wet–dry cycle testing.

Under natural conditions, soil moisture migration occurs in an approximately one-dimensional manner due to factors such as evaporation, rainfall infiltration, and fluctuating groundwater levels [39]. Soil cracking patterns are influenced by gradients in moisture content, which in turn are affected by varying moisture migration mechanisms, consequently impacting the rate of soil strength degradation. To emulate the natural impact of dry–wet cycles on soil strength, the sample’s lateral surfaces were enveloped in cling film, thereby confining moisture migration to the sample’s top and bottom during the wetting and drying processes [40]. A water film transfer technique was employed for wetting, while a low-temperature (40 °C) oven facilitated the drying phase. Initially, filter paper and permeable stones were positioned on the sample’s top and bottom surfaces, respectively, followed by the placement of a sponge saturated with distilled water atop the sample. Subsequently, a 10 mL syringe was utilized to uniformly distribute distilled water until the desired moisture content was achieved, followed by immediate weighing. The sample was then placed in a controlled humidity chamber for a 24 h equilibration period. For drying, a constant temperature forced-air oven was used, and the sample was periodically weighed until the target moisture content was attained, followed by a 24 h stabilization in a controlled humidity chamber. These procedures were iteratively executed to simulate varying numbers of dry–wet cycles. Figure 4 presents a schematic outline of the soil sample’s wetting procedure.

In this study, the dry–wet cycling test for improved soil is categorized into two distinct groups: the dry density group and the cycling path group. In the former, the focus is solely on variations in dry density, with the lower limit of the dry–wet cycles and the amplitude of moisture content held constant. Conversely, the latter group considers the lower limit of dry–wet cycles and the amplitude of moisture content, while maintaining a constant dry density.

(1)Dry density group

In this study, the dry–wet cycle group was structured with specific dry densities of 1.5 g/cm^3^, 1.6 g/cm^3^, 1.7 g/cm^3^, and 1.8 g/cm^3^. Additionally, the number of dry–wet cycles (n) was established at 0, 1, 2, 3, 4, 5, 6, 9, and 12 iterations. Unconsolidated undrained (UU) triaxial shear tests were performed under confining pressures of 80 kPa, 100 kPa, and 150 kPa, resulting in a cumulative total of 108 specimens. These values were selected to simulate the range of in situ stresses that the soil is likely to experience in real-world conditions. Our preliminary site investigations indicated that these values were representative of the depths and loading conditions the soil would be exposed to. Our choice was also informed by the relevant literature in the field. Past research involving similar soil types and conditions has frequently utilized these pressure ranges, making it valuable for comparative analysis. Table 6 delineates the experimental design for the dry density group. The soil’s natural moisture content during the summer dry season was employed as the lower limit moisture content (wd = 5.5%) for these dry–wet cycle tests. During the wetting phase, it was observed that each specimen maintained approximately 88% saturation until a constant weight was attained. Consequently, the upper limit moisture content was established at an 88% saturation level for each specified dry density.

(2)Cycle path group

In the cycle path group, the dry density of the specimens was maintained at a consistent 1.7 g/cm^3^. Five distinct dry–wet cycle paths were defined: 6.5–24.5%, 5.55–11.5%, 5.5–17.5%, 9.5–21.5%, and 13.5–25.5%, with corresponding cycle amplitudes of 18%, 6%, 12%, 12%, and 12%. The subgroups within this main group were designated as Group ①, ②, ③, ④, and ⑤. Comparative analysis between Groups ② and ③ permits the investigation of the upper limit water content, while a similar analysis involving Groups ③, ④, and ⑤ enables the evaluation of the lower limit water content. Additionally, Groups ①, ②, and ③ serve to examine the influence of dry–wet cycle amplitude. Table 7 outlines the experimental scheme for the cycle path group.

#### 2.2.2. Observation of Microscopic Pore and Particle Structures

Scanning Electron Microscopy (SEM) produces high-resolution images of sample surfaces, allowing researchers to observe and analyze microscopic structures and features. When analyzing SEM images, one typically follows these steps: (1) pre-processing, utilizing image processing software (like ImageJ 2.0 or Photoshop 7.0) to eliminate background noise, specks, or other interferences, adjusting contrast and brightness, and ensuring details and features within the image are clearly discernible; (2) identification and categorization of features, differentiating and classifying various structures and attributes in the image based on shape, size, and arrangement patterns, and using annotation tools to mark areas or features of interest; (3) quantitative analysis, measuring the dimensions of features, such as particle size, length, width, etc.; analyzing particle morphology and distribution, and, when applicable, conducting elemental or compositional analysis on specific regions (e.g., using Energy-Dispersive X-ray Spectroscopy (EDS)) (This instrument is produced by Nanjing Nanguang Geological Instrument Co., Ltd. in Nanjing. China). In essence, the aim of SEM image analysis is to furnish a profound understanding of the sample surface characteristics, providing either qualitative or quantitative data for further studies or applications.

At the microscale, soil’s fundamental attributes predominantly relate to its pore and particle types, encompassing aspects like arrangement, roughness, shape, contact, bonding, erosion, stagnation, and migration. Utilizing Scanning Electron Microscopy (SEM) and digital image processing, the particle structures were meticulously observed. The SEM micrographs underwent preprocessing via the Image-Pro Plus 6.0 software, which included tasks such as filtering, correction, and binarization (refer to Figure 5). Geometrical data, including pore and particle counts, perimeter, and area, were discerned and extracted. This enabled a comprehensive quantitative assessment of the evolution in pore and particle shape, size, and arrangement [40].

Geometric data are procured to perform a quantitative assessment of the dynamic evolution in pore and particle morphology, arrangement, and size [40]. Subsequent to each wetting and drying cycle, soil samples are extracted and sectioned into cubic specimens measuring 3 cm × 3 cm × 3 cm, centered within each sample. These are further refined into 1 cm × 1 cm × 1 cm cubes utilizing a microtome. Following natural air-drying, the sample is cleaved along its sedimentation axis to expose a fresh surface, which is subsequently gold-coated in preparation for Scanning Electron Microscopy (SEM) analysis of the pore and particle structures [41].

## 3. Macroscale and Microscale Effects of Strength Degradation in Improved Soil

### 3.1. Effect of Dry–Wet Cycles on the Strength of Improved Soil

Figure 6 delineates the shear strength evolution of the improved soil subject in response to varying confining pressures across multiple dry–wet cycles. The data indicate a marked degradation in soil shear strength as the number of dry–wet cycles escalates. This degradation is further accentuated under increased confining pressure, manifesting overtly non-linear characteristics at a higher pressure of 100 kPa. Notably, the most pronounced reductions in shear strength occur within the initial four cycles, transitioning to a phase of gradual strength diminution between the fourth and sixth cycles. At a confining pressure of 100 kPa, the soil’s shear strength decreases by 34.85% and 39.4% after four and six cycles, respectively, before plateauing into a stable state. Thus, beyond six cycles, the shear strength exhibits relative stability, marking the upper boundary of the dry–wet cycle’s impact on soil degradation. Prior studies have posited that a large extent of strength attenuation in various types of soils—such as expansive soil, compacted loess, and sliding zone soil—occurs primarily within the initial three cycles [40,42,43]. In contrast, the presence of fibers in the improved soil in this study amplified the soil’s resilience, rendering it more durable against the adverse effects of repeated dry–wet cycles.

### 3.2. Effect of Dry–Wet Cycles on the Microstructure of the Improved Soil

For an exhaustive analysis of the improved soil’s microstructure, Scanning Electron Microscopy (SEM) images with a 500× resolution were employed for qualitative assessment of the skeletal structure. These were juxtaposed with images captured at different magnifications. Figure 7 exhibits SEM images representing varying numbers of dry–wet cycles affecting the improved soil. It is evident from Figure 7 that with an escalating number of dry–wet cycles, the larger soil aggregates progressively disintegrate into smaller units. Notably, the particle contact modes predominantly transition from face-to-face and edge-to-plane to edge-to-face and point-to-plane configurations [44]. Repeated dry–wet cycles facilitate reciprocal moisture migration within the soil, subjecting soil pores—the conduits for this moisture transfer—to continual washout. This leads to the mobilization of fine soil debris and soluble cementing material, resulting in pore enlargement and aggregate disintegration. Concurrently, the reduction of readily soluble minerals triggers particle suspension, diminishing the contact area between soil particles. To furnish a detailed quantitative analysis, SEM images with 1000× magnification were processed using Image-Pro Plus 6.0 software for calibration, contrast and gamma adjustment, as well as edge processing and binarization. This analysis focused on the ramifications of dry–wet cycles on the average pore diameter, particle abundance, and morphological dimensions.

#### 3.2.1. Average Pore Diameter

Referring to the pore division standard of Shear et al. (2023) and the pore classification method for compacted silt obtained by Ren et al. (2019), a compacted mercury test, the pores of the improved soil are divided into four categories: large pore size (>60 μm), medium pore size (60~5 μm), small pore size (5~0.1 μm), and micropore size (<0.1 μm). In the field of geotechnical engineering, the dimension of soil micropores is commonly quantified in terms of their diameters. Nevertheless, it is critical to acknowledge that the actual pore geometries are heterogeneous. To address this variability, the Image-Pro Plus 6.0 software employs the concept of an equivalent circle, using its diameter to represent the actual pore area S t when calculating the average pore diameter Dd. The computational formula employed is as follows:(1)$$D_{d}=\sqrt{\frac{4S}{\pi }}$$

Drawing upon the pore classification criteria delineated by Shear et al. [43] and Ren et al.’s 2019 methodology for categorizing consolidated fine-grained soil based on mercury intrusion porosimetry tests, we identify four distinct categories of pores in the improved soil: large pores (exceeding 60 µm), medium pores (ranging from 60 to 5 µm), small pores (spanning 5 to 0.1 µm), and micropores (less than 0.1 µm). Specifically, large pores predominantly occur as interaggregate pores, medium pores primarily serve as intra-aggregate pores, small pores are chiefly interparticle pores, and micropores are mainly intraparticle pores.

Figure 8 delineates the correlation between the distribution of four distinct pore sizes and the number of wet–dry cycles. A medium-sized pore is 60-5 um. Small-sized pores are 50-0.1 um. Initially, in the natural state (n = 0), small- and medium-sized pores predominate, constituting 37.5% and 32.2% of the total pore volume, respectively, as compared to the pore volume of compacted clay ascertained using mercury intrusion porosimetry by Ren et al. [44]. The main hydration products, such as hydrated calcium silicate and hydrated calcium aluminate, generated by internal hydration and volcanic ash reactions in the improved soil possess particle diameters below 0.001 µm on the nanoscale [45]. These products augment the microparticle count, thereby elevating the volume percentage of small pores. As the wet–dry cycles continue, Figure 8 shows a negative correlation between micropore size and cycle number, with the most significant change occurring during the inaugural cycle. Concurrently, medium and large pores display a positive correlation with cycle number. This suggests that as wet–dry cycles progress, micropores not only expand but also become interconnected, thereby diminishing the relative volume percentages of medium and large pores.

#### 3.2.2. Particle Abundance

The morphological characteristics of soil particles play a pivotal role in the physical properties of the soil, significantly influencing its fluidity, aeration, permeability, and susceptibility to erosion. Among these, the “Particle Richness” (also known as “Granulometric Abundance” or “Particle Morphology Index”), typically denoted as “C”, stands out as a key metric to describe these characteristics. The Particle Richness (C) primarily characterizes the shape, size, and distribution of soil particles. This metric provides invaluable insights into the coarseness, uniformity, and implications on the mobility of moisture and nutrients within the soil. To enhance the precision in assessing the soil particle morphology affected by wet–dry cycles, a particle elongation index, denoted as C, has been introduced. This index represents the ratio of the short axis (B) to the long axis (L) of an individual soil particle. The C-value ranges between 0 and 1; a lower C-value indicates a more elongated particle shape, while a C-value approaching 1 signifies a more circular shape. The formula for calculating C is presented below:(2)$$C=\frac{B}{L}$$

Figure 9 elucidates the correlation between particle abundance and the number of wet–dry cycles. Under natural conditions (n = 0), the particle abundance in the improved soil predominantly falls within the 0.5-0.8 range, indicating a preponderance of circular or elliptical particles. The absence of particles with abundance values between 0.1 and 0.2 at (n = 0) is noteworthy; however, these particles emerge after one wet–dry cycle. This emergence is likely due to the erosion of larger, irregular particles caused by the reciprocating migration of water within the soil pores. Following three cycles of wet–dry alternation, the particle abundance peaks within the 0.5–0.7 range. For values of 0.1, 0.2, 0.8, and 0.9, this peak occurs after nine cycles. Although the particle abundance exhibits complex variations, it generally increases with the number of wet–dry cycles. Consequently, initially rough or elongated particles gradually adopt more rounded, flattened, or nearly spherical shapes. Concurrently, while water-induced erosion disaggregates larger particles into smaller, irregular forms, new aggregates form due to the coalescence of these smaller particles, accounting for the complex changes observed in particle abundance.

#### 3.2.3. Fractal Dimension of Morphological Distribution

The fractal dimension is a mathematical concept employed to depict the intricacies and patterns of complex geometries. It serves as an effective metric to quantify the intricacy and irregularity of soil particle shapes or the porosity of soil structures. The configuration and distribution of soil particles play pivotal roles in influencing the soil’s physical, chemical, and biological properties. When applied to soil particles or their porous structures, the fractal dimension offers a nuanced means to measure the complexity of soil’s micro-architecture. For instance, rough and irregular soil particles tend to have a higher fractal dimension, while smoother, more uniformly shaped particles typically exhibit a lower value. Fractal theory, which originated in the 1970s, offers a quantitative framework for describing the micro-properties of rock and soil that contain irregular particles and pores. Extensive empirical data have substantiated that the pore structures of rock and soil exhibit distinct fractal characteristics, thereby furnishing a robust theoretical foundation for and methodological approach to the scrutiny of rock and soil particulate units as well as pore structures [46,47,48]. Utilizing the morphological distribution fractal dimension for equivalent area and perimeter, as formulated by Moore and Donaldson [49], this study assesses the roughness of the soil particle units. The calculation formula for this dimension is as follows:(3)$$\mathrm{Log}(\mathrm{Perimeter})=\frac{D}{2}\times \mathrm{Log}(\mathrm{Area})+c$$

In this equation, D represents the fractal dimension of soil particles, while c serves as the fitting constant.

Figure 10 illustrates the relationship curve for the fractal dimension of soil particles under varying numbers of dry–wet cycles. In its natural state (n = 0), the fractal dimension of the improved soil is 1.522. However, after undergoing nine dry–wet cycles, this value declines to 1.427. Notably, the fractal dimension demonstrates a decreasing trend throughout the dry–wet cycle process, with the most significant reduction—2.56%—occurring after three cycles.

The primary factor contributing to the observed reduction in particle fractal dimension is the cyclic migration of moisture within the soil pores during dry–wet cycles. This process not only erodes large particles but also dissolves soluble minerals, leading to a progressive breakdown of the original soil particles into varying sizes. Specifically, as moisture circulates within the fine pores, it continues to decompose and abrade the constituent soil particles, resulting in a more diversified particle size distribution. Consequently, the fractal dimension of the soil particles diminishes over successive dry–wet cycles, signifying a corresponding decline in the mechanical strength of the improved soil.

### 3.3. Investigation of Macroscale and Microscale Mechanisms

Under the influence of successive dry–wet cycles, the shear strength of the improved soil exhibits a continuous decline, governed by a complex interplay of microscopic mechanisms. This deterioration can be attributed to the intricate synergy of water–material–chemical–mechanical interactions within the soil matrix. Specifically, clay minerals play a pivotal role, as their hydrophilic nature critically influences the extent of pore contraction and the corresponding reduction in soil strength. Given that repeated dry–wet cycles translate microstructural damage into macroscopic strength degradation, it becomes imperative to investigate the underlying mechanisms at both macroscopic and microscopic levels [40].

During the soil wetting process, water molecules permeate the soil matrix and adhere to hydrophilic clay minerals, forming an expanded layer of bound water film. This process triggers the swelling of clay minerals and the consequent dilation of the adjacent pore spaces. Simultaneously, a double electric layer forms and disperses between the proximate clay mineral regions and the surrounding aqueous solution, with concentration gradients accumulating continuously. Subject to pressure differentials, water molecules consequently displace the surrounding particles, creating new pore spaces. In addition, repetitive pore water migration results in the dissolution of intergranular binding substances, weakening interparticle cohesion and accelerating soil deterioration. During the drying phase, water loss occurs in a tiered fashion: the free water in larger pores is lost first, followed by the weakly bound water distal to the clay minerals, and ultimately, the crystalline water layer and surrounding bound water are depleted [50]. This sequential loss triggers clay mineral shrinkage and releases the corresponding crystalline-layer water. The cyclical wetting and drying processes induce pore and particle edge erosion, culminating in increased pore size and particle abundance. When microlevel damage accrues to a critical extent, microscopic cracks emerge, which manifest macroscopically as a decline in soil strength [51].

Based on the analysis of the aforementioned experimental results, the structural deterioration of improved soil under dry–wet cycles has been comprehensively characterized, as illustrated in Figure 11. Additionally, the mechanisms contributing to the decline in strength at both microscopic and macroscopic levels are summarized as follows:

The iterative process of wet–dry cycles induces various microstructural changes in the improved soil. Specifically, the cycles contribute to the reciprocal dissolution of the binding substances in the micropores, as well as the cyclical expansion and contraction of clay minerals. These changes lead to hydrolysis of the binding substances and enlargement of the soil pores, thereby causing fatigue damage at the microstructural level and a corresponding deterioration in macroscopic strength. The weakening of the soil’s macroscopic strength can be attributed to the interplay between the expansion and dissolution potentials, which act repeatedly on the soil’s microstructure, resulting in irreversible fatigue damage [52]. According to the dialectic relationship between the microscale and macroscale, alterations at the microstructural level significantly impact the soil’s macroscopic deformation behavior. The macroscopic mechanical behavior, in turn, is governed by its evolutionary law [53]. Therefore, the decline in shear strength of improved soil under wet–dry cycles is a cumulative result of multiple microstructural changes, including alterations in particles, binding substances, and pores [43].

The specific mechanisms leading to pore enlargement in the studied context primarily stem from the increased dissolution rates of certain soluble constituents in the soil matrix. As these constituents dissolve, the voids they leave behind coalesce to form larger pores [54,55]. Moreover, cyclic loading and unloading from environmental factors can cause expansion and contraction, facilitating the growth of existing pores. Regarding particle disaggregation, several processes come into play. Firstly, the repeated wetting and drying cycles cause volumetric changes in the soil aggregates, leading to stress development between particle contacts. Over time and with repeated cycles, these stresses weaken the bonds holding the particles together, resulting in disaggregation. Additionally, the freeze–thaw process in some environments can cause ice lens formation, exerting pressure and leading to further disaggregation of particles.

Under the effects of wet–dry cycles, there are distinct differences in the micro–macro characteristics and degradation in strength between fiber-reinforced soil and unreinforced soil:

Micro-Macro Characteristics:(1)Pore Structure: Fiber reinforcement can enhance soil cohesion, reducing changes in the pore structure caused by wet–dry cycles. In contrast, unreinforced soil may experience more significant alterations in its pore structure due to moisture absorption and release.(2)Crack Formation: Unreinforced soil is more prone to crack formation, especially on the surface, under wet–dry cycles. The presence of fibers in reinforced soil can effectively prevent or minimize the progression of cracks.(3)Particle Re-arrangement: The re-arrangement of particles in fiber-reinforced soil may be inhibited by the fibers, leading to increased stability. On the other hand, unreinforced soil might exhibit greater tendencies for particle re-arrangement and structural collapse.

Degradation in Strength:(1)Compressive Strength: After wet–dry cycles, fiber-reinforced soil typically retains a higher compressive strength due to the added tensile and flexural capacities from the fibers. The compressive strength of unreinforced soil might decrease significantly after such cycles.(2)Shear Strength: The introduction of fibers can enhance the shear strength of the soil, especially after wet–dry cycles. In contrast, the shear strength of unreinforced soil may reduce due to the effects of moisture.(3)Erosion Resistance: Fiber-reinforced soil generally demonstrates better erosion resistance, as fibers can mitigate the erosive action of water on the soil. Unreinforced soil might be more susceptible to erosion.

Other Factors:(1)Water Retention: The incorporation of fibers might influence the soil’s water retention properties, which could further impact the effects of wet–dry cycles on the soil’s characteristics.(2)Biological Activity: Reinforcement with fibers could potentially affect the microbial activity in the soil, further influencing soil stability.

In summary, fiber reinforcement can significantly improve soil performance and stability under wet–dry cycles, particularly in terms of its micro–macro characteristics and strength degradation. However, these outcomes also depend on various factors such as fiber type, length, shape, and the amount of reinforcement.

## 4. Conclusions

The study embarks on a comprehensive exploration into both the macroscopic and microscopic facets that influence strength degradation in silty soil once improved by regenerated polyester fiber. The methodology employed hinges on dry–wet cycle tests complemented by Scanning Electron Microscopy (SEM) analyses. Delving deeper into the research outcomes, we discern several pivotal findings:(1)It is evident that the shear resistance of the enhanced soil undergoes notable degradation, correlating directly with the increase in the number of dry–wet cycles. Intriguingly, the application of higher confining pressures acts as a countermeasure to this degradation, revealing a trend wherein the shear resistance retains more of its original strength. Moreover, under these escalated confining pressures, the shear resistance degradation trajectory showcases non-linearity, hinting at possible complex underlying micro-interactions.(2)With the progression of dry–wet cycles, a transition in the particle interaction modes is discerned in the enhanced soil. This evolution predominantly sways from face-to-face contact, gradually gravitating toward edge-to-face and point-to-face configurations. Parallelly, an intriguing development is observed at the microscopic level where the soil’s micropores not only expand but also increasingly interconnect. This trend nudges mesopores and macropores out of their spaces, leading to a decreased volume fraction. Noteworthy is the surge in the abundance of soil particles with every subsequent dry–wet cycle. Contrastingly, the particle fractal dimension wanes, underscoring the progressive weakening of the soil’s inherent mechanical strength.(3)The dry–wet cyclic exposure incites a rhythmic expansion and contraction within the clay minerals. This dynamic, in tandem with the recurring dissolution of bonding agents by the micropore water present in the treated soil, orchestrates a series of fatigue-induced damages at the microstructural layer. More specifically, hydrolysis reactions targeting the bonding agents and a consistent enlargement of pores are triggered. These micro-changes collectively culminate in a tangible compromise of the macroscopic mechanical strength of the treated soil, warranting further attention in soil stabilization endeavors.

## Figures and Tables

**Figure 1 polymers-15-04367-f001:**
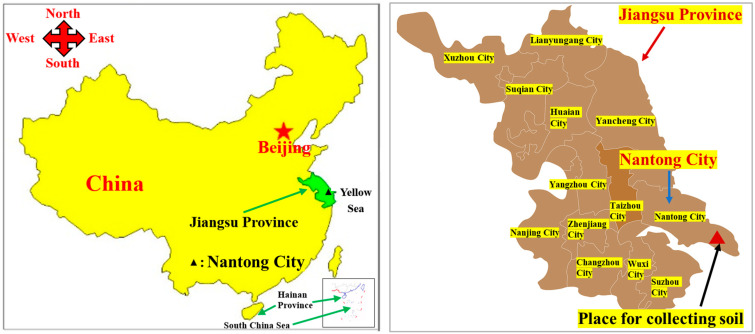
Location of the soil texture.

**Figure 2 polymers-15-04367-f002:**
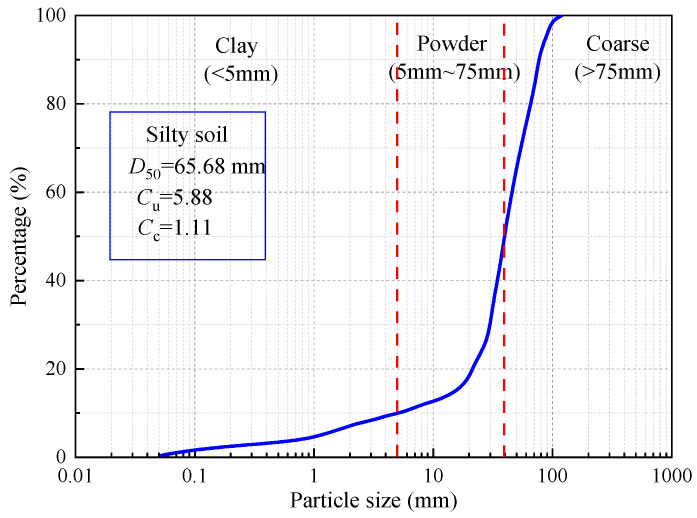
Particle size distribution curve of the tested soil sample.

**Figure 3 polymers-15-04367-f003:**
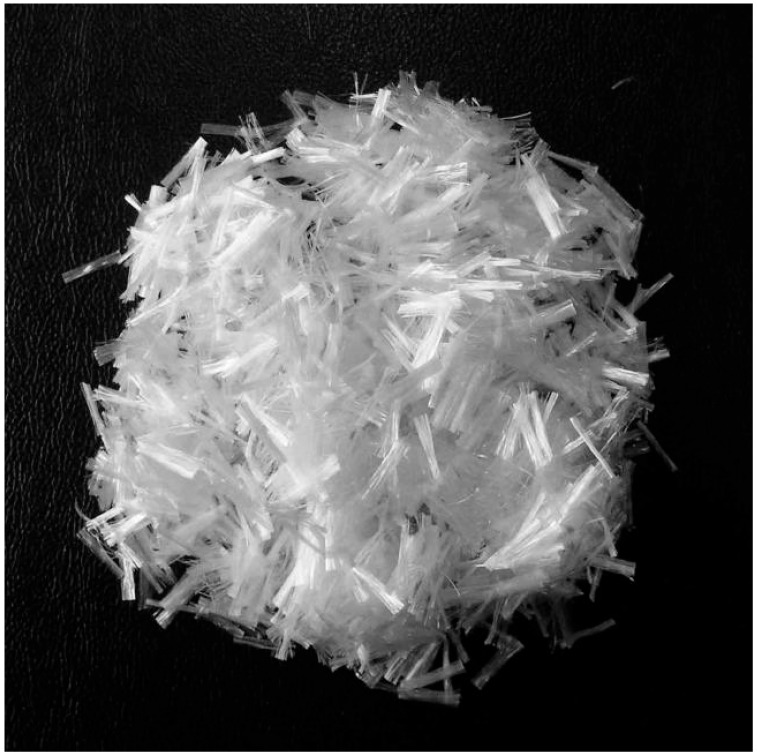
Renewable polyester fiber.

**Figure 4 polymers-15-04367-f004:**
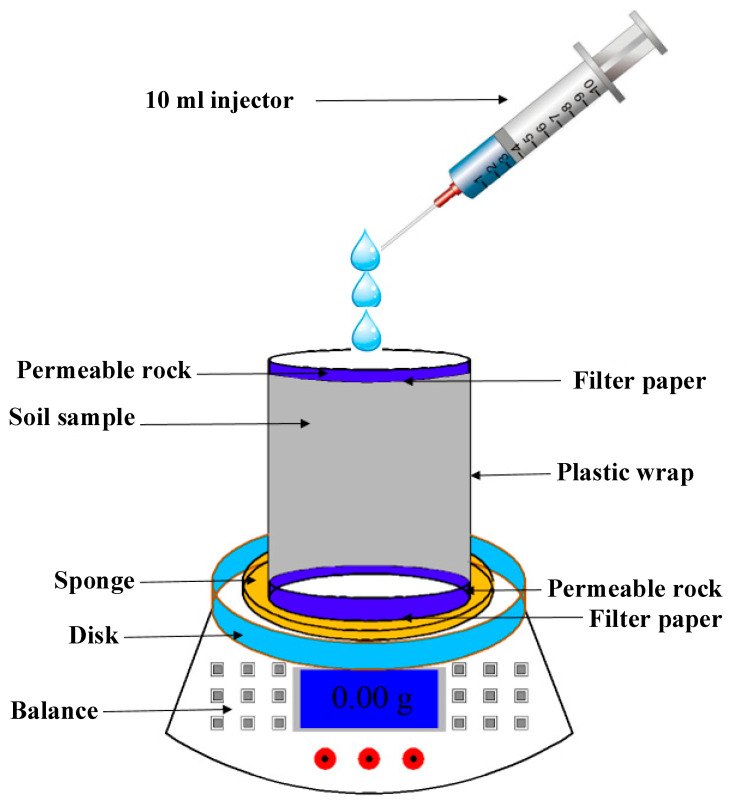
Schematic diagram of soil sample wetting process.

**Figure 5 polymers-15-04367-f005:**
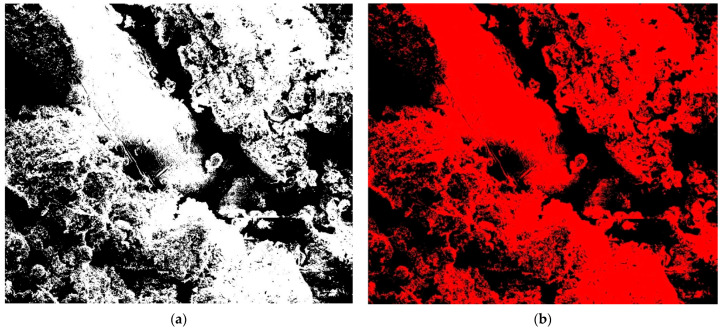
Pre-processing of SEM images: (**a**) image binarization and (**b**) particle structure recognition.

**Figure 6 polymers-15-04367-f006:**
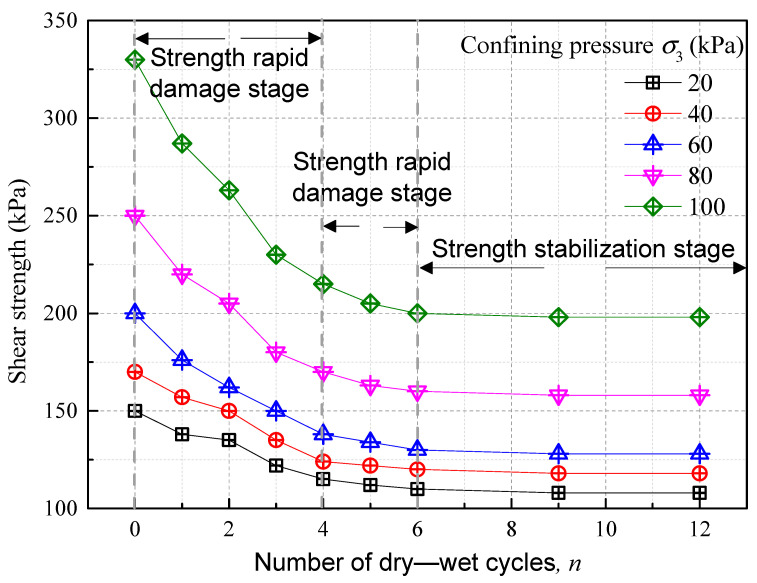
Improved soil shear strength evolution curve with dry–wet cycle times.

**Figure 7 polymers-15-04367-f007:**
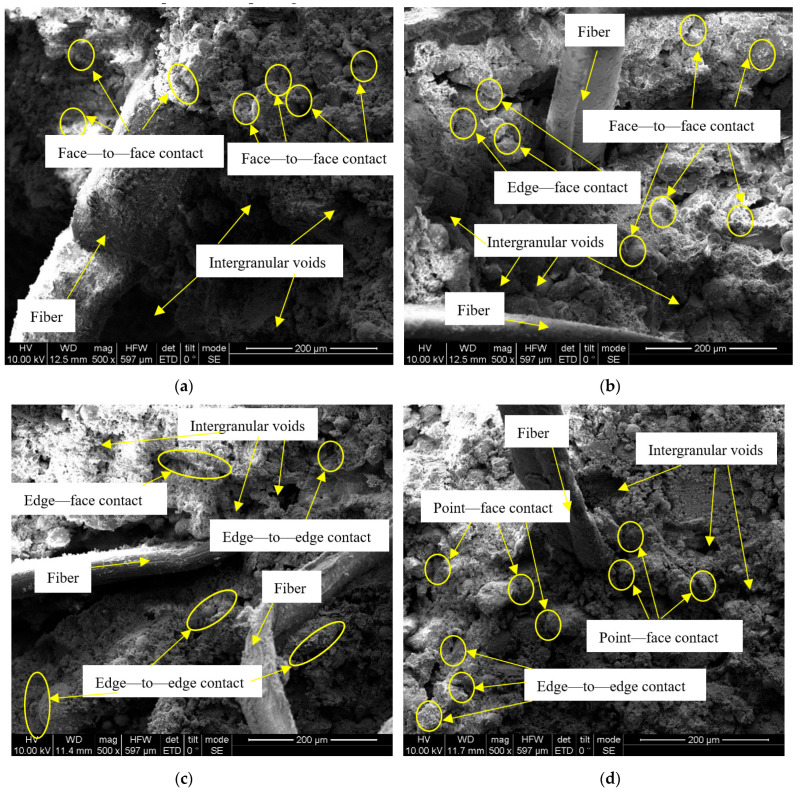
SEM images of improved soil under different dry–wet cycles: (**a**) zero times; (**b**) three times; (**c**) six times; (**d**) nine times.

**Figure 8 polymers-15-04367-f008:**
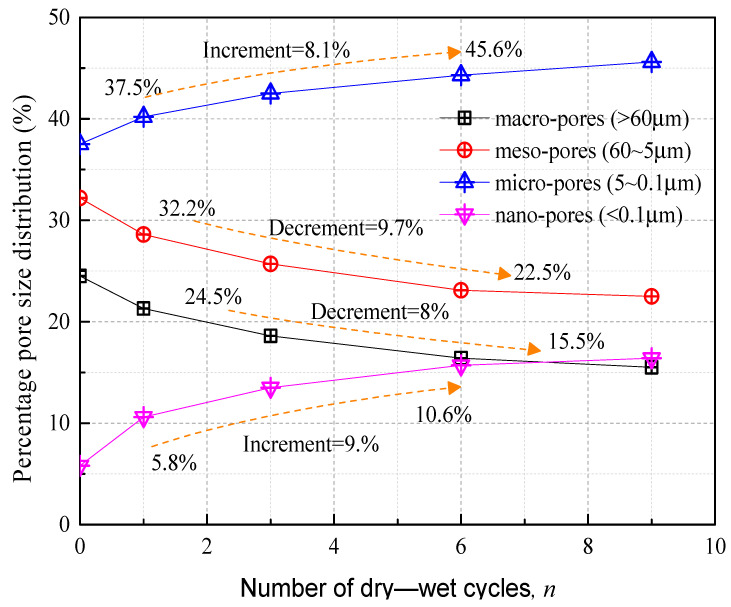
Relationship curve between the percentage of four pore sizes and the number of wet–dry cycles.

**Figure 9 polymers-15-04367-f009:**
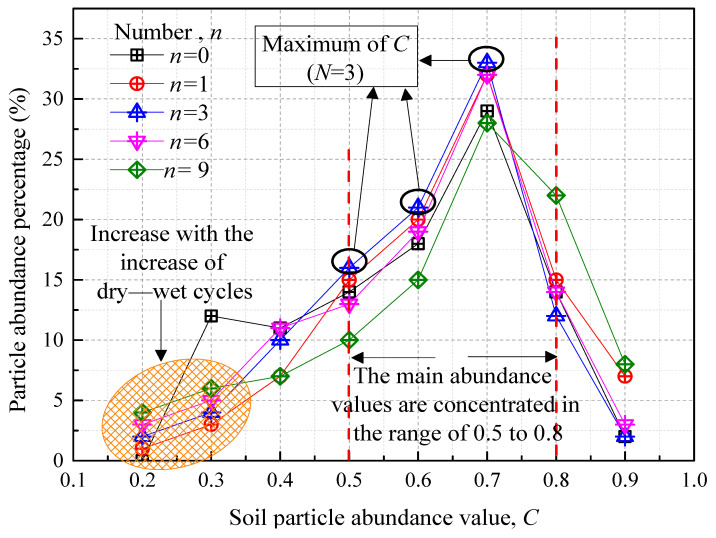
Soil particle abundance versus dry–wet cycle number relationship curve.

**Figure 10 polymers-15-04367-f010:**
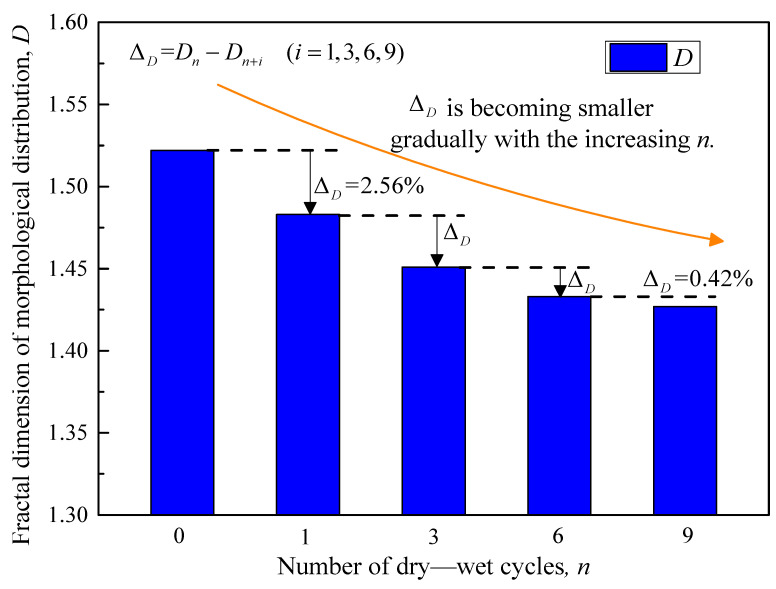
The variation of the morphological distribution fractal dimension.

**Figure 11 polymers-15-04367-f011:**
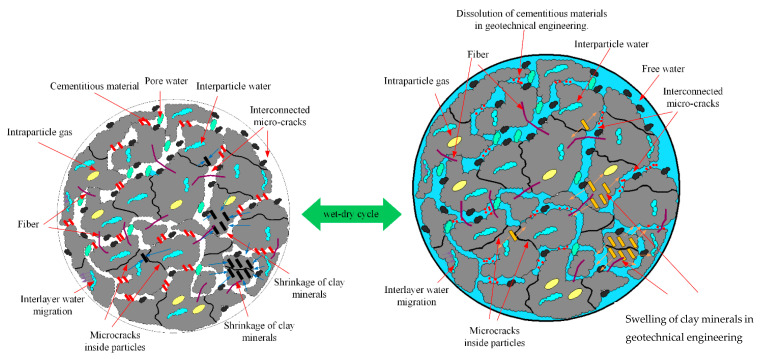
Schematic diagram of the structural evolution of modified soil under dry–wet cycles.

**Table 1 polymers-15-04367-t001:** Main physical properties of the tested soil sample.

Natural Moisture Content/%	*LL*/%	*PL*/%	*I_P_*	Particle Size Distribution			Maximum Dry Density /(g/cm^3^)	Optimum Moisture Content/%	Specific Gravity /(g/cm^3^)	pH
				<5 μm	5–75 μm	>75 μm				
25.4	31.6	22.8	8.8	11.3	79.8	8.9	1.81	16.45	2.71	8.21

**Table 2 polymers-15-04367-t002:** Chemical composition analysis results of the soil sample.

Chemical Composition	SiO_2_	Al_2_O_3_	CaO	Fe_2_O_3_	K_2_O	MgO	Na_2_O	SO_3_	P_2_O_5_	Other	Firing Loss
Content/%	61.32	13.24	6.68	3.41	2.61	2.47	2.17	0.23	0.18	2.01	5.68

**Table 3 polymers-15-04367-t003:** Basic physical properties of quicklime.

Specific Gravity /(g/cm^3^)	pH	Clay Content/% (<2 μm)	Silt Content/% (2 μm~75 μm)	Sand Content/% (>75 μm)
3.31	12.4	5.4	42.7	51.9

**Table 4 polymers-15-04367-t004:** Chemical analysis of quicklime.

Chemical Composition	CaO	SiO_2_	Al_2_O_3_	Fe_2_O_3_	MgO	SO_3_	Na_2_O	K_2_O	TiO_2_	SrO	MnO	Firing Loss
Content/%	65.23	2.62	1.16	0.74	0.46	0.13	0.20	0.18	0.053	0.029	0.028	24.36

**Table 5 polymers-15-04367-t005:** Fly ash major chemical composition and performance indicators.

Main Chemical Composition (%)				Specific Gravity	Optimal Moisture Content (%)	Maximum Dry Density (g/cm^3^)
CaO	Fe_2_O_3_	Al_2_O_3_	SiO_2_	2.15	23.2	1.34
2.8	7.9	28.4	46.2			

**Table 6 polymers-15-04367-t006:** Experimental design for the dry density group.

Number	Cycle Number	Dry Density	Cycle Path	Number	Cycle Number	Dry Density	Cycle Path
1	0	1.5	6.5 ~ 24.5%	1	0	1.6	6.5 ~ 24.5%
2	1	1.5		2	1	1.6	
3	2	1.5		3	2	1.6	
4	3	1.5		4	3	1.6	
5	4	1.5		5	4	1.6	
6	5	1.5		6	5	1.6	
7	6	1.5		7	6	1.6	
8	9	1.5		8	9	1.6	
9	12	1.5		9	12	1.6	
1	0	1.7	6.5 ~ 24.5%	1	0	1.8	6.5 ~ 24.5%
2	1	1.7		2	1	1.8	
3	2	1.7		3	2	1.8	
4	3	1.7		4	3	1.8	
5	4	1.7		5	4	1.8	
6	5	1.7		6	5	1.8	
7	6	1.7		7	6	1.8	
8	9	1.7		8	9	1.8	
9	12	1.7		9	12	1.8	

**Table 7 polymers-15-04367-t007:** Experimental design for the cycle path group.

Number	Cycle Number	Cycle Path	Number	Cycle Number	Cycle Path	Number	Cycle Number	Cycle Path	Number	Cycle Number	Cycle Path	Number	Cycle Number	Cycle Path
①	0	6.5 ~ 24.5%	②	0	5.5 ~ 11.5%	③	0	5.5 ~ 11.5%	④	0	9.5 ~ 21.5%	⑤	0	13.5 ~ 25.5%
①	1		②	1		③	1		④	1		⑤	1	
①	2		②	2		③	2		④	2		⑤	2	
①	3		②	3		③	3		④	3		⑤	3	
①	4		②	4		③	4		④	4		⑤	4	
①	5		②	5		③	5		④	5		⑤	5	

## Data Availability

The data are contained within the article.

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
